# Adipose Tissue: A Source of Stem Cells with Potential for Regenerative Therapies for Wound Healing

**DOI:** 10.3390/jcm9072161

**Published:** 2020-07-08

**Authors:** Lucy V Trevor, Kirsten Riches-Suman, Ajay L Mahajan, M Julie Thornton

**Affiliations:** 1Plastic Surgery and Burns Research Unit, University of Bradford, Bradford BD71DP, UK; lucy.trevor@doctors.org.uk (L.V.T.); Ajay.Mahajan@bthft.nhs.uk (A.L.M.); 2Department of Plastic and Reconstructive Surgery, Bradford Teaching Hospitals NHS Foundation Trust, Bradford BD71DP, UK; 3Centre for Skin Sciences, Faculty of Life Sciences, University of Bradford, Bradford BD71DP, UK; k.riches@bradford.ac.uk

**Keywords:** adipose tissue, fat grafting, stromal vascular fraction, adipose-derived stem cells, regenerative medicine, wound healing

## Abstract

Interest in adipose tissue is fast becoming a focus of research after many years of being considered as a simple connective tissue. It is becoming increasingly apparent that adipose tissue contains a number of diverse cell types, including adipose-derived stem cells (ASCs) with the potential to differentiate into a number of cell lineages, and thus has significant potential for developing therapies for regenerative medicine. Currently, there is no gold standard treatment for scars and impaired wound healing continues to be a challenge faced by clinicians worldwide. This review describes the current understanding of the origin, different types, anatomical location, and genetics of adipose tissue before discussing the properties of ASCs and their promising applications for tissue engineering, scarring, and wound healing.

## 1. Introduction

The obesity epidemic has transformed the way we think about adipose tissue. Adipose tissue, once considered as a simple inert tissue functioning as insulation and energy storage, is now seen as one of the largest endocrine organs in the body, containing a variety of cell types, with multiple potential applications for regenerative medicine. Just one-third of adipose tissue volume is comprised of mature adipocytes; the remainder consists of a heterogenous group of preadipocytes, fibroblasts, mesenchymal stem cells (MSCs), endothelial progenitor cells (EPCs), pericytes, macrophages, T-cells, and erythrocytes in a stromal vascular cell network (Figure 1) [1]. Enzymatic digestion of adipose tissue produces a heterogeneous population of precursor cells within a cell pellet which is referred to as the stromal vascular fraction (SVF). Components of the SVF were quantified by the International Federation for Adipose Therapeutics and Science (IFATS) and International Society for Cellular Therapy (ISCT) who reported cells were present in the following proportions: 15–30% stromal cells, 10–20% endothelial cells, 10–15% lymphocytes, 10–15% granulocytes, 5–15% monocytes, 3–5% pericytes, and <0.1% stem and progenitor cells [2]. This diverse SVF cell population, derived from adipose tissue, has been investigated for its potential therapeutic use in regenerative medicine. More specifically, MSCs present in the SVF, termed adipose-derived stem cells (ASCs), have been explored as the key cell type responsible for the beneficial effects of SVF. ASCs have been shown in vitro to promote human dermal fibroblast proliferation and migration and production of collagen by both direct cell-to-cell contact and paracrine activation [3]. It remains to be seen whether these observed beneficial effects may be harnessed into suitable therapy options for chronic and nonhealing wounds. This paper aims to explore the characteristics of adipose tissue and its role in wound healing in order to look towards the potential therapeutic implications of this diverse tissue.

## 2. Adipose Tissue: Origins, Types, Distribution, and Genetics

Adipose tissue, like muscle and bone, is generally regarded as having a mesodermal origin. However, stem cells derived from the mouse neural crest have been reported to be able to differentiate into adipocytes in culture, indicating that craniofacial adipose depots may originate from the neural crest [4]. The emerging view is that the origins of adipose tissue are complex and in-depth lineage tracing has yet to be fully explored.

Adipose tissue depots are classically separated into two distinct types: white adipose tissue (WAT) and brown adipose tissue (BAT) [5]. Adipocyte morphology in WAT and BAT is distinctly different and gives rise to their diverse roles. WAT contains adipocytes with a single large lipid droplet that accounts for >90% cell volume, whereas BAT adipocytes contain numerous smaller lipid compartments and abundant mitochondria giving rise to the brown appearance after which they are named [5,6]. The white adipocytes predominantly function to store energy as triglycerides and provide thermal insulation and physical cushioning of the body, whereas brown multilocular adipocytes are active in the dissipation of energy through the production of heat [6].

WAT is the predominant form of adipose tissue in the human body whereas BAT is much less common and can be found in the supraclavicular, periadrenal, and paravertebral regions of the body [1]. Anatomical sites in which WAT develops are called ‘depots’; one example of a depot is the visceral white adipose tissue (vWAT) which surrounds human internal organs [7]. WAT associated with the skin has been further described as two separate populations by Driskell et al. in 2014. They suggest that dermal white adipose tissue (dWAT) describes a depot in the dermis underlying the reticular dermis whereas subcutaneous white adipose tissue (sWAT) defines a unique and separate population with no anatomical division in humans but is situated below striated muscle (the *panniculus carnosus*) in mice [7]. As anatomical separation of the two populations is poorly defined in humans and these two populations were, until recently, considered synonymous, there is little evidence of the differences in the behavior between these two populations [8]. The subcutaneous adipose layer varies in thickness and is determined by multiple variables such as sex, age, body site, and nutritional status, and is completely absent in certain body sites, namely, the eyelids, clitoris, and penis [9].

Major human depots include vWAT in the omentum, intestines, and perirenal areas, and sWAT in the buttocks, thighs, and abdomen (Figure 2). WAT can be found in other less apparent areas such as the retro-orbital space and within the bone marrow [10]. The anatomical distribution of adipose tissue is believed to influence the occurrence of metabolic disease. An increased central distribution of adipose tissue (apple-shaped) is thought to promote a higher risk of metabolic diseases, such as cardiovascular disease and type two diabetes, whereas increased deposits of subcutaneous adipose tissue in the thighs and hips (pear-shaped) have been shown to have little or no risk associated with metabolic disease [11]. Different anatomical depots are responsive to different steroids; for example, the breast and thighs are influenced by gonadocorticoids, whereas the neck and upper back are influenced by glucocorticoids. This is observed in Cushing’s syndrome; the “buffalo-hump” is a characteristic sign of an excess of the glucocorticoid cortisol [10]. Twin and population studies in humans have shown that genetics account for 30-–70% of the variability seen in adipose tissue distribution, independent of BMI, with the remaining influences believed to be unique environmental effects and age [12]. One example of heritable control of adipose tissue distribution is observed in southern Africa in women of Khoikhoi ethnicity; they display steatopygia [13]. This accumulation of large amounts of fat on the buttocks gives a curvaceous figure which is a modern characteristic of feminine beauty often requested in aesthetic augmentation procedures and may be achieved by gluteal fat grafting.

## 3. Wound Healing

Effective wound healing is essential to restore the continuity of the skin barrier following damage. Cutaneous wound healing is a complex and dynamic process which is dependent on numerous coordinated interactions between cells, growth factors, and extracellular molecules. This intricate series of events attempts to restore the integrity, function, and tensile strength of an injured tissue. Wound healing is classically divided into three sequential but overlapping stages: inflammation, proliferation, and remodeling [14]. Hemostasis begins immediately after tissue injury and is achieved by vasoconstriction, platelet aggregation, and the formation of a fibrin clot, through activation of the coagulation cascade [15]. The fibrin clot provides a provisional matrix for the migration of inflammatory cells to the wound site [15]. The inflammatory phase prepares the wound environment for healing by removing debris and bacteria from the wound to prevent infection, much like the purpose of surgical debridement. Proinflammatory cytokines are released by macrophages including platelet-derived growth factor (PDGF) which stimulate the migration of dermal fibroblasts to the wound [15]. The proliferative stage of wound healing is composed of the synthesis of tissue granulation, contraction, and re-epithelialization. Migrating fibroblasts recruited to the wound bed produce extracellular matrix (ECM) proteins and growth factors. Their movement into the wound is assisted by their secretion of ECM-cleaving matrix metalloproteinases (MMPs) [15]. Macrophages provide a source of growth factors required for angiogenesis and new capillaries formed give the stroma a pink granular appearance; this granulation tissue functions as a scaffold for dermal fibroblast migration [16]. The provisional ECM is constructed of fibrin, fibronectin, and hyaluronic acid which are gradually replaced by collagens [15,16]. Some fibroblasts assume a myofibroblast phenotype distinguished by the presence of alpha smooth muscle actin (α-SMA) myofilaments which allow contraction of the wound [16,17]. Transforming growth factor beta (TGFβ) signaling facilitates myofibroblast contraction and ECM deposition in wound healing and scar formation [7]. Once abundant collagen has been deposited into the granulation tissue, many fibroblasts in the wound undergo apoptosis [16,18]. Physical connections between keratinocytes named desmosomes are broken and they no longer adhere to one another, allowing migration of epidermal keratinocytes laterally across the wound behind which epidermal keratinocytes proliferate [16]. These keratinocytes migrate across the wound bed until the epithelial tongues meet and seal the wound [18]. During the remodeling phase, which can last up to one year, the wound matures by the simultaneous formation and degradation of collagen fibers [19]. Disorganized type III collagen is remodeled into aligned type I fibers in order to more closely resemble the structure of surrounding tissue. This is achieved by the action of MMPs, which are endopeptidases secreted by fibroblasts, macrophages, and endothelial cells [20]. Water is resorbed so collagen fibers lie closer and cross-link, which reduces scar thickness and increases its tensile strength. Nevertheless, despite these changes, the tensile strength of the formed scar tissue will only reach approximately 80% of the tensile strength of uninjured skin [19]. Remodeling requires a stable balance between MMPs and tissue inhibitors of matrix metalloproteinases (TIMPs). Excessive accumulation of disorganized fibrous molecules results in fibrosis, which impairs skin function and appearance [21]. Due to the central role of growth factors and cytokines in the regulation of wound healing, abnormal growth factor or cytokine expression may negatively affect this coordinated process leading to chronic wound healing and excessive scarring. Numerous factors such as advancing age, metabolic disorders, and environmental influences may disrupt the normal wound healing process and result in inadequate healing or pathological dermal scarring such as hypertrophic/keloids scarring. These present clinical burdens to which management could potentially be improved through the use of regenerative medicine techniques.

## 4. Adipose Tissue as a Source of Stem Cells

Human MSCs are adult stem cells with the potential to differentiate into multiple cell linages of mesenchymal tissues [22]. Minimal criteria to define MSC were proposed by the Mesenchymal and Tissue Stem Cell Committee of the ISCT in 2006. These criteria have been described in order to generate standardization in comparing research observed by different investigators. According to these criteria, in order for a cell to be considered an MSC, it must demonstrate plastic adherence in standard culture conditions. Cells must express the cell surface markers CD105, CD73, CD90, and a lack of expression of the cell surface markers CD45, CD34, CD14 or CD11b, CD79α or CD19, and HLA-DR. Finally, they must have the potential to differentiate into osteoblast, adipocyte, and chondroblast cell linages in vitro [23]. A single unique surface marker to characterize ASCs has not been identified and therefore the ISCT criteria are still used to identify ASCs. A small core of stem cell transcription factors, namely, octamer-binding transcription factor 4 (OCT4), sex determining region Y-box 2 (SOX2), and nanog homeobox (NANOG), are essential for maintaining the pluripotency of embryonic stem cells [24]. A recent study has demonstrated that the knockdown of *NANOG* in human ASCs cultured from sWAT triggers a loss of pluripotency, demonstrating that expression of this transcription factor is also essential for maintaining the properties of human ASC in culture [25].

The term “preadipocyte” has been used by researchers to describe a progenitor cell population that is adipose lineage-committed, whereas ASCs are defined as multipotent stem cells with osteogenic, adipogenic, and chondrogenic potential [1]. Both populations are present in WAT and can be isolated from the SVF of subcutaneous adipose tissue. In contrast to bone marrow, adipose tissue is more abundant and accessible, and MSCs can be harvested through liposuction procedures under local anaesthetic [26]. The prevalence of obesity is on the rise in the Western world and elective liposuction procedures are performed frequently. ASCs have been shown to have a longer culture period and higher proliferative capacity than bone marrow MSCs [27]. Adipose tissue therefore represents a more favourable source of MSCs than bone marrow due to these qualities.

Peroxisome proliferator-activated receptor gamma (PPARγ) is the central regulator for differentiation of progenitors into adipocytes in all adipose tissue depots [28]. PPARγ is a nuclear receptor of which there are two isoforms: PPARγ1 and PPARγ2. As well as this important role in adipogenesis, PPARγ along with CCAAT-enhancer-binding protein alpha (C/EBPα) appear to be dually required for the maintenance of terminally differentiated adipocytes as knockout of PPARγ and C/EBPα in mouse studies in vivo leads to widespread death of adipocytes and reduction in sWAT [29]. PPARγ2 is a key component of adipogenesis and maintenance, and expression in humans has been shown to be variable in ASCs isolated from different depots [30].

The various anatomical depots of adipose tissue vary in volume and composition and therefore it is reasonable to postulate that the ASCs situated in different depots may also have variable characteristics. It has been identified that adipose donor site impacts on the characteristics of the ASCs harvested. One study revealed the proliferative rate of ASCs derived from the abdomen, limb, and knee was significantly higher than that of ASCs derived from the thigh and hip of human donors. In contrast, osteogenic characteristics were shown to be increased in the less proliferative hip and thigh ASCs [31]. Another study compared ASCs derived from different human adipose depots including: the upper arm, medial thigh, trochanteric, and both superficial and deep (below *scarpas fascia*) abdominal. ASCs isolated from superficial abdominal adipose tissue have been shown to be less susceptible to apoptotic stimuli than those isolated from these other sites [30]. The survival of ASCs following grafting is one of the key challenges in regenerative ASC grafting techniques, therefore the identification of this characteristic of abdominal ASCs makes this depot a favorable location for harvesting. Another study also showed that human ASCs derived from abdominal subcutaneous adipose tissue had more advantageous properties. They demonstrated higher rates of proliferation and increased adipogenic potential in ASCs that were derived from abdominal subcutaneous adipose tissue than in those from abdominal visceral adipose tissue [32]. It appears both ASC phenotypes vary with both depth and location and therefore are important to consider in the selection of an appropriate anatomical site for ASC harvest. In addition to the perceived advantages of harvesting ASCs from the superficial abdominal anatomical depot, liposuction of the abdomen is a popular site choice for surgeons due to its accessibility, abundance, and aesthetic improvements for the patient.

As well as anatomical site, another variant shown to impact on ASC properties is donor age. Human ASCs derived from younger patients were shown to proliferate at higher rates and have an enhanced ability to differentiate into mature adipocytes than those isolated from older patients [30], while ASCs harvested from infants had enhanced osteogenic and angiogenic capabilities compared to those from older age groups [33]. Better characterization of donor phenotype will enable us to better utilize the appropriate characteristics to achieve the desired effects of cell-based therapies.

Liposuction is a minimally invasive cosmetic procedure which involves aspirating subcutaneous fat using a cannula under negative pressure. Liposuction aspirate primarily consist of saline and local anesthetic solution, blood, and adipose tissue fragments which separate into fluid and fatty portions. Cells isolated from the fatty portion of this suspension are termed processed lipoaspirate (PLA) and have been the predominant source of ASCs for investigation [34]. However, it has been suggested that the fluid portion of liposuction aspirated, termed liposuction aspirate fluid (LAF), also contains ASCs [35]. ASCs can also be harvested from surgical excision of subcutaneous adipose tissue which ensures a purer sample of adipose tissue is obtained.

Rodbell (1966) was the pioneer of a procedure to isolate SVF cells from adipose tissue using collagenase digestion [36]. This method has since been adapted over many years and published methods for isolating ASCs from adipose tissue involve the same core steps. Protocols begin with mincing the adipose tissue to increase surface area before incubating with collagenase solution in order to digest the tissue. Collagenase digestion is stopped after a period of incubation by adding an equal volume of culture medium containing 10% fetal bovine serum. The resultant digestion medium is then filtered to remove tissue debris before being centrifuged (700G for 10 min). This separates the contents into three distinct fractions of floating mature adipocytes, a middle aqueous fraction, and a reddish cellular pellet containing SVF cells at the bottom. The other contents are discarded to leave the SVF pellet. Erythrocytes are a major contaminant present in the SVF pellet and may be lysed to isolate a purer population using an ammonium-chloride-potassium lysis buffer. The buffer is then diluted by the addition of an equal volume of media before repeating centrifugation. The cell pellet is resuspended in media and the cells are seeded into dishes/plates and incubated at 37 °C to allow attachment [37]. This step allows for selection of the plastic adherent population of cells which includes ASCs. Mincing of fat may be done by hand but may be omitted from the methodology if lipoaspirates are used, provided the cannula dimensions are small enough to provide finely minced tissue fragments [26]. Liposuction therefore reduces the time taken to harvest cells, although if ultrasound-assisted liposuction is used, the number of ASC cells isolated and their proliferative capacity are reduced [38].

## 5. Tissue Engineering Techniques for Wound Healing

Precursor cells from adipose tissue are of great interest within the field of reconstructive surgery. However, as described previously, isolation techniques for human ASCs for in vitro studies currently include collagenase digestion. Collagenase is derived from a bacterial source (*Clostridium histolyticum*), and its use is currently prohibited for the manipulation of fat grafts for human use. The US Food and Drug Administration (FDA) regulations state that fat grafts must be minimally manipulated, enzyme-free, and used in the same surgical procedure as harvesting [39]. Consequently, nonenzymatic methods using mechanical forces to loosen the structural integrity of adipose tissue extracellular matrix (ECM), such as centrifugation, have been used in human clinical trials to purify liposuction aspirates. In 2008, Yoshimura et al. reported their novel method of supplementing fat grafts with simultaneously extracted SVF cells in a process termed cell-assisted lipotransfer (CAL) for cosmetic breast surgery and reported an increase in breast circumference in centrifuged fat grafts [35]. Beneficial effects have since been reported from the transplant of lipoaspirates containing ASCs into chronic wounds caused by oncologic radiotherapy in human clinical trials [40]. Poorly healing burn wounds have also been shown to respond to novel adipose tissue treatment of fat grafting around and beneath the wound bed in an Indian tertiary burns center. This large study of 60 patients showed statistically significant improvements in measurable outcomes of wound healing such as granulation tissue production, epithelialization, and reduction in wound surface area and was observed in patients treated with autologous fat grafts for the treatment of three-week-old burn wounds that had previously shown no signs of healing [41]. Fat grafting has been repeatedly reported to reduce pain and enhance cosmetic appearance of scars by improving color, elasticity, and thickness [42]. However, some of the reported outcomes observed following fat grafting may be due to technique and needle insertion causing release of fibrotic tissue that improves scar maturation. The benefits of improvements observed in skin cannot be fully attributed to ASCs due to restrictions preventing solitary isolation of ASCs for human use and procedural impacts on tissue. It is therefore important for ASCs to be studied in vitro in order to control the many variables and clearly establish mechanisms of action.

One proposed mechanism for the action of ASCs is through their secretome (see Table 1). ASCs secrete cytokines, growth factors, adipokines, neurotrophic factors, and angiogenic factors such as PDGF, fibroblast growth factor (FGF), vascular endothelial growth factor (VEGF), hepatocyte growth factor (HGF), and angiopoietin into their secretome [43]. This diverse mix of various components exert their beneficial effect through their paracrine activity and can be harnessed by the use of conditioned media (CM) to deliver this to target cells. Human ASCs have been shown through qPCR and Western blot techniques to express FGF-2 and it has been demonstrated by ELISA that they secrete FGF-2. Both expression and secretion were more pronounced when they were cultured as spheroids [44]. FGF-2 has an important role in cutaneous wound healing in promoting fibroblast proliferation and migration via the extracellular signal-regulated kinase (ERK)1/2 and c-Jun N-terminal kinase (JNK) pathways [45,46]. One study quantified the concentration of FGF-2 and HGF in cultured media generated from commercially acquired human ASCs by ELISA and found that there was an 18-fold greater concentration of FGF-2 than HGF in the secretome [47]. As a potent component of CM with known beneficial influences on fibroblasts, it may contribute to the favorable effects of ASC CM in wound healing. The adipokine, adiponectin, has been identified in the proteome of human ASCs through two-dimensional gel electrophoresis–tandem mass spectroscopy and Western immunoblot [48]. In one study, it was demonstrated by qPCR that human neonatal dermal fibroblasts express receptors for adiponectin and these cultures were shown by ELISA to increase production of collagen I in response to adiponectin at a concentration of above 1 µg/mL in a dose-dependent manner [49]. It therefore appears that adiponectin secreted by ASCs has a role in upregulating matrix production by dermal fibroblasts and may have a role in wound healing (Figure 3).

ASC CM has been shown to improve properties of wound healing in dermal fibroblasts when observed in vitro. One study showed that human keloid-derived fibroblast proliferation was reduced following treatment with human ASC CM in a BrdU proliferation assay [50]. This suggests that ASCs may downregulate proliferation of diseased fibroblasts causing problematic scarring via excessive ECM deposition in wounded skin. Cooper et al. demonstrated treatment with ASC CM increased dermal fibroblast migration in a scratch wound assay by 43% [45].

In addition to soluble factors, extracellular vesicles are secreted by ASCs serving as vectors for the transfer of nonsoluble components including proteins, lipids, messenger RNA (mRNA), microRNA (miRNA), and long noncoding RNA (lncRNA). These can all contribute to the paracrine signaling of ASCs and may also influence wound healing [45]. Extracellular vesicles range from 30 to 100 nm in size and originate from endosomes in the cytoplasm, or from the plasma membrane, and are an important mode of intercellular communication. Once released, these vesicles can then either fuse directly with the plasma membrane of the target cell or enter the target cell by endocytosis where they can exert a regulatory function [51]. Two main types of extracellular vesicles have been described on the basis of their size, exosomes, and microvesicles. Exosomes are small membrane-bound lipid vesicles, approximately 30–100 nm in diameter [52]. In comparison, microvesicles are generally larger extracellular vesicles, ranging from 50 to 1000 nm in diameter [52]. Extracellular vesicles transfer a cargo of non-soluble components, including RNA, from one cell to another. Furthermore, mRNA contained in these vesicles can be translated into proteins by the recipient cell. Messenger RNA is single-stranded RNA, which can be translated into a protein molecule within the target cell. Messenger RNA vector-mediated transgene expression has a rapid and transient effect on cells and is one mechanism by which cells exert their effects through paracrine signaling [53]. In contrast, miRNAs are single-stranded RNA molecules that influence the post-transcriptional regulation of gene expression by binding specific targets on mRNAs, downregulating their expression and thereby ultimately shaping the cell transcriptome [54]. Long noncoding RNAs are RNA strands that exceed 200 nucleotides in length, exhibiting a wide range of sizes, shapes, and functions. They are described as noncoding as they are not translated into proteins, although recent studies have demonstrated their involvement in different aspects of gene regulation by various mechanisms of action [55]. Exosomes derived from human ACSs have been shown to accelerate wound healing by promoting the migration, proliferation, and collagen synthesis of human dermal fibroblasts [45,56]. Microvesicles secreted by human ASCs have also been shown to enhance the proliferation and migration of human foreskin fibroblasts in vitro [57]. It therefore appears that both exosomes and microvesicles secreted by ASCs have a key role in wound healing, however, further clarification as to their unique composition has yet to be illustrated [52].

The paracrine activity of ASCs has been extensively researched to date, although there remains potential to harness other desirable characteristics of these cells for regenerative medicine techniques. In vivo transplantation of ACS in mice revealed that engrafted cells displayed keratinocyte markers and released keratinocyte growth factor (KGF), providing evidence in support of the potential ability to differentiate into keratinocytes [58]. Also, in the same study, ASCs were able to differentiate into cells expressing endothelial cell markers, providing evidence for the differentiation of ASCs into endothelial cells to aid neovascularization [58]. Another in vivo study demonstrated that labeled human ASCs seeded into silk fibroin–chitosan scaffolds and implanted into full-thickness murine skin defects significantly increased wound closure [59]. This study demonstrated that engrafted ACSs displayed expression of epidermal epithelial cell markers at four weeks post-op, thus suggesting that wound closure was aided by the differentiation of ASCs into epidermal cells [59]. It is well established that ASCs have the capability to differentiate into osteogenic, adipogenic, and chondrogenic lineages, a characteristic which is fundamental to their unique identity [1]. However, the hypothesis that ASCs may aid wound healing through differentiation is still poorly explored.

## 6. Further Adipose-Derived Targets with Implications for Wound Healing

Despite the mounting evidence in favor of ASCs as the principal component of adipose responsible for improved wound healing, it appears that additional cell types may also have a role which must be explored further. In murine skin, immature adipocytes have been shown to be activated during the proliferative phase of wound healing, and that mature adipocytes and fibroblasts appear in healing wounds concurrently [62]. PPARγ inhibitors prevent the occurrence of this observation and therefore it appears the population of adipocytes present in the wound bed occurs via adipogenesis [62]. Human mature adipocytes, cultured in 3D spheroids, have been shown to stimulate the differentiation of fibroblasts into myofibroblasts through the secretion of paracrine factors. Adipocyte CM increased fibroblast migration, contractility, α-SMA expression and collagen production when compared to pre-adipocyte CM [63]. Furthermore, endothelial progenitor cells isolated from the SVF appear to have a role in neovascularization during the proliferative phase of wound healing. Endothelial progenitor cells have been cultured from the SVF of adipose tissue and have demonstrated the ability to form capillary-like structures in three-dimensional scaffolds [64].

Fibroblasts are another one of the many cell types which comprise the SVF. Given dermal fibroblasts are the key cell type involved in extracellular matrix formation and remodeling, it is important to distinguish the role of this subpopulation of fibroblasts as they have the potential to demonstrate similar properties. Fibroblasts isolated from the dermo–hypodermal junction have been shown to have distinct characteristics including a lower proliferative rate and a reduced capacity for collagen lattice contraction when compared to intermediate reticular fibroblasts and superficial papillary fibroblasts [65]. Fibroblasts isolated from the dermo–hypodermal junction have also demonstrated the capacity for three-lineage mesenchymal differentiation, giving them an MSC-like cellular identity. Despite these similarities to MSCs and their anatomical proximity to subcutaneous adipose tissue within the hypodermis, hierarchical clustering based on transcriptome profiling has shown they still exhibit a clear “fibroblast” molecular identity that is quite distinct from MSCs [65]. It therefore appears that fibroblasts from the dermal–hypodermal junction have a separate role to both ASCs and reticular or papillary dermal fibroblasts in wound healing due to their unique properties.

In contrast to the apparent positive role of adipose tissue during wound healing, a detrimental effect of obesity on wound healing has long been established. This converse relationship may be in part due to the composition of adipose in obese individuals. Obesity induces adipocyte hypertrophy and hyperplasia, which impairs the metabolic function of adipocytes [66]. The rate of angiogenesis in obese individuals does not increase proportionally to meet the rate of adipocyte growth, and consequently, a relative hypoxia is also likely to contribute to poorer wound healing [66]. Another paradox observed in obese individuals is a reduced concentration of the adipose-derived cytokine adiponectin [66]. Adiponectin is a potent mediator in the regulation of cutaneous wound healing and has been shown to enhance mouse keratinocyte proliferation and migration in vitro in a dose-dependent manner through ERK activation [67]. It is important to understand these limitations of adipose tissue in order to ensure new techniques benefit patient outcomes rather than increase the complications encountered.

## 7. Future Outlook

There is currently no gold standard treatment for scars, and wound healing is a challenge faced by clinicians across the globe. Regenerative techniques exploiting the benefits of ASCs are still in the primitive stages of development, however, promising therapeutic potential has been shown by the pioneers of autologous fat grafting. Fat grafting has revealed improvements of both function and form in patients with abnormal wound healing and scarring, while in vitro studies have shown that ASCs isolated from human adipose tissue may be responsible for these effects. Limitations, owing to the restrictions in place by the FDA, on the manipulation of adipose tissue prior to transplantation, in addition to the difficulties associated with viability and graft-take post-fat graft transplantation, have led to the need to utilize alternative mechanisms for ASC regenerative therapies. The ASC secretome, composed of soluble factors and extracellular vesicles, has therefore been explored as a resolution to these obstructions which may enable clinicians to harness their therapeutic potential for improved wound healing. In order to explore the significance of findings from in vitro studies supporting the ASC secretome as a stimulant for proliferation, migration, and matrix remodeling, in vivo wound healing studies are also required. Future research will help to explore this large gap in understanding from bench to bedside and allow for new breakthroughs in this up-and-coming specialty of regenerative medicine.

## Figures and Tables

**Figure 1 jcm-09-02161-f001:**
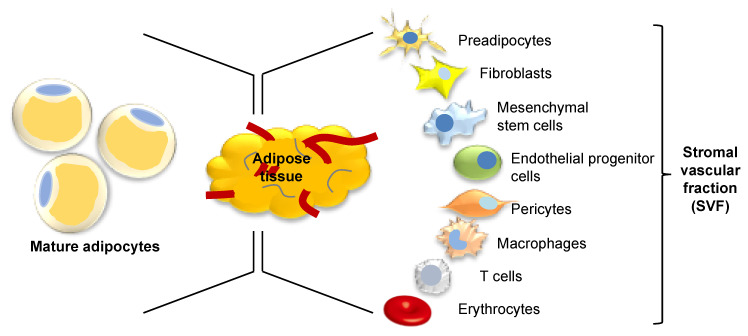
Composition of adipose tissue. Adipose tissue comprises a complex mixture of cells, one-third of which are mature adipocytes. Following enzymatic digestion, the remaining two-thirds contains multiple cell types known as the stromal vascular fraction.

**Figure 2 jcm-09-02161-f002:**
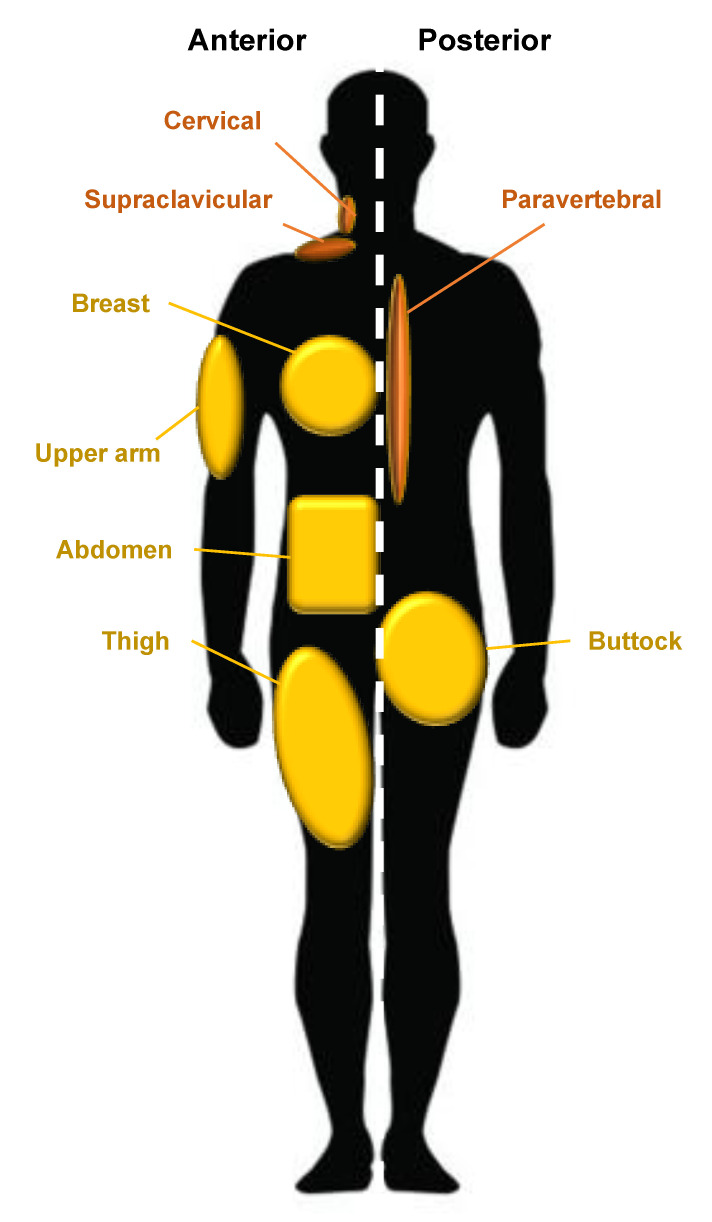
Adipose tissue distribution in humans. WAT is widely distributed around the body; principal WAT depots are illustrated in gold and BAT depots in brown. Silhouette indicates the anterior and posterior anatomical position on the left and right, respectively.

**Figure 3 jcm-09-02161-f003:**
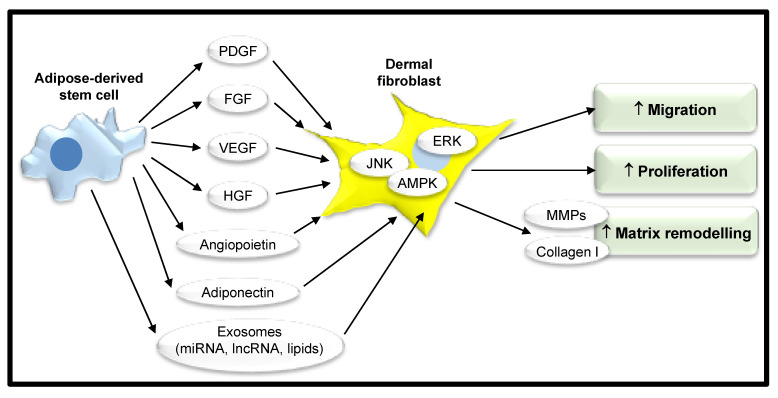
Adipose-derived stem cells promote wound healing. Adipose-derived stem cells have a complex secretome that includes, for example, growth factors and adipokines. These induce migration, proliferation, and matrix remodeling in neighboring dermal fibroblasts through various intracellular signaling pathways to enhance wound healing. PDGF, platelet-derived growth factor; FGF, fibroblast growth factor; VEGF, vascular endothelial growth factor; HGF, hepatocyte growth factor; ERK, extracellular signal-regulated kinase; JNK, c-Jun N-terminal kinase; AMPK, adenosine monophosphate protein kinase; MMPs, matrix metalloproteinases; miRNA, microRNA; lncRNA, long noncoding RNA.

**Table 1 jcm-09-02161-t001:** The modulation of dermal fibroblast function by the ASC secretome: implications for improved wound healing.

Stimulatory Effect of ASC Secretome on Dermal Fibroblast Function	Paracrine Factors Secreted by ASCs	References
Increased proliferation	FGF-2	[45,46,60]
	HGF	[60]
	Exosomes	[56,61]
	Microvesicles	[57]
Increased migration	FGF-2	[45,46]
	Exosomes	[45,56,61]
	Microvesicles	[57]
Matrix Remodeling (increased production of collagen I and III, TGFβ, FGF2, and MMP1)	Adiponectin	[49]
	Exosomes	[56,61]

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
