# Peer review of "Adipose Tissue: A Source of Stem Cells with Potential for Regenerative Therapies for Wound Healing"

_jcm, 2020, doi:10.3390/jcm9072161_

Round 1
Reviewer 1 Report
Trevor et al. reviews the role of adipose-derived stem cells (ASCs) and their potential in scars and impaired wound healing. The focus of this review lays the current understanding of the origin, different types, anatomical location and genetics of adipose tissue before discussing the properties of ASCs and their promising applications for tissue engineering, scarring and wound healing.
Figure 3 is somehow ‘very simple’ and could be adapted by giving more information on factors influencing ECM. Moreover, the addition of an additional table including references on the newest finding on migration, proliferation and ECM formation would round up the review.
Finally, there are many papers dealing with the role AD-MSC and donor site (see reference below) and their role in differentiation. The authors shall take up the discussion and extend their view towards the specific role in wound healing of various Ad-MSCs.
Reumann et al. Donor Site Location Is Critical for Proliferation, Stem Cell Capacity, and Osteogenic Differentiation of Adipose Mesenchymal Stem/Stromal Cells: Implications for Bone Tissue Engineering. Int. J. Mol. Sci. 2018, 19, 1868; doi:10.3390/ijms19071868
Author Response
Thank you for this feedback which has been extremely helpful in improving our manuscript. We include a point-by-point response below with all changes highlighted in yellow.
In response to reviewer 1:
Lines 217-218
We have updated Figure 3 and added some additional information to show the different components of the secretome of ASCs and their effect on dermal fibroblasts. The figure legend has also been updated accordingly.
Lines 300-301
We have composed a table (Table 1) as requested containing the latest findings on proliferation, migration and extracellular matrix formation by dermal fibroblasts in response to ASCs. We have added some extra referenes (40,41,42)
Lines 119-122
We have included the suggested Reumann et al. reference in the discussion surrounding ASCs and donor site (reference 19)
Lines 185-190
We have also added some extra text to discuss the impact of donor age on ASC characteristics including one extra reference to add to this discussion (reference 21).
In response to reviewer 2:
Lines 100-104
We have included extra text to discuss the suggested paper by Pitrone et al. (reference 15), plus one other reference has been added for clarity (reference 14).
In response to reviewer 3:
There were no further suggestions, but we thank reviewer 3 for their very positive comments
Additional changes:
Lines 85-86
Figure 2 has been moved to be embedded within the relevant text rather than be located at the end of the paper.
Reviewer 2 Report
Good and concise review on the promising use of Adipose Stem Cells (ASC) to be used for hard-to-heal wounds, keloids and burns. Please add a recent paper of Pitrone et al. (Int J Mol Sci. 2019) that regards ASC transcriptions factors and imaging of cell cultures.
To date, unfortunately, we must be satisfied with in vitro studies that allow us to hope to soon have the opportunity to use ASCs for clinical purposes.
Good discussion on today's liposuction methods as a source of ASC.The conclusions are concrete. I express a favorable opinion on thepublication of this interesting review, useful for the reader interested in this field of research.
Author Response

(The authors gave the same response as above.)

Reviewer 3 Report
- Well written and referenced summary article. No suggestions from my view. Accomplishes its goal and is clear.
Author Response

(The authors gave the same response as above.)

Round 2
Reviewer 1 Report
no further comments